# A Selective, Dual Emission β-Alanine Aminopeptidase Activated Fluorescent Probe for the Detection of *Pseudomonas aeruginosa*, *Burkholderia cepacia*, and *Serratia marcescens*

**DOI:** 10.3390/molecules24193550

**Published:** 2019-09-30

**Authors:** Linda Váradi, Elias Y. Najib, David E. Hibbs, John D. Perry, Paul W. Groundwater

**Affiliations:** 1The University of Sydney School of Pharmacy, Camperdown Campus, Sydney, NSW 2006, Australia; elias.najib@sydney.edu.au (E.Y.N.); david.hibbs@sydney.edu.au (D.E.H.); paul.groundwater@sydney.edu.au (P.W.G.); 2Microbiology Department, Freeman Hospital, High Heaton, Newcastle upon Tyne NE7 7DN, UK; John.Perry@nuth.nhs.uk

**Keywords:** fluorescent probe, *Pseudomonas aeruginosa*, bacterial detection, self-immolative, 3-hydroxyflavone

## Abstract

Selective detection of β-alanyl aminopeptidase (BAP)-producing *Pseudomonas aeruginosa*, *Serratia marcescens*, and *Burkholderia cepacia* was achieved by employing the blue-to-yellow fluorescent transition of a BAP-specific enzyme substrate, 3-hydroxy-2-(*p*-dimethylaminophenyl)flavone derivative, incorporating a self-immolative linker to β-alanine. Upon cellular uptake and accumulation of the substrate by viable bacterial colonies, blue fluorescence was generated, while hydrolysis of the *N*-terminal peptide bond by BAP resulted in the elimination of the self-immolative linker and the restoration of the original fluorescence of the flavone derivative.

## 1. Introduction

Early and specific detection of pathogenic bacteria is essential for informed clinical decision making and the selection of the appropriate therapeutic choices [1,2]. Colourimetric/fluorogenic enzyme substrates are acknowledged as gold standards and are often applied in clinical settings [3,4]. They incorporate a small coloured/fluorescent heterocyclic molecule covalently attached to an amino acid or sugar moiety that can be selectively metabolized by the bacterium of interest [5]. By tailoring the metabolic target (e.g., a sugar or amino acid) moiety within the substrate, selectivity and specificity can be achieved. β-Alanyl aminopeptidase (BAP) is specifically expressed by *Pseudumonas aeruginosa*, *Burkholderia cepacia*, and *Serrratia marcescens*, and can be targeted by β-alanyl moieties as the *N*-terminal fragments of peptide-like derivatives. *P. aeruginosa* is a multidrug resistant respiratory pathogen posing significant risk for immunocompromised patients [6], *B. cepacia* can cause severe lung infections in cystic fibrosis patients [7], while *S. marcescens* is a notorious causative agent of outbreaks in neonatal intensive care units (NICUs) [8].

Strain-specific aminopeptidase activity can be targeted by incorporating *N*-terminal amino acids as enzyme recognition moieties. For example, use of the colourimetric BAP substrates, 7-amino-1-pentyl-3H-phenoxazin-3-one **1** (1-pentylresorufamine [PRF]) (Scheme 1) [9], can result in up to 99.9% sensitivity; however, the time taken to obtain reliable results can be 18–72 h. Fluorogenic enzyme substrates possess inherently enhanced detection sensitivity (as a result of displaying emitted light against a non-emitting background [10]) when compared to chromogenic alternatives, which can result in reduced diagnostic timeframes. Conjugation of *N*-terminal enzyme recognition moieties to hydroxylic chromophores/fluorophores can be achieved by incorporating a self-immolative linker between the chromophore and the amino acid [11]. For example, a BAP substrate incorporating a coumarin derivative **2** (Scheme 1) in liquid growth media enabled a diagnostic result within 6 h [12]. A large proportion of the previously published fluorogenic enzyme substrates detect metabolic activity based on an off-to-on fluorescent signal in the blue wavelength region [12,13], however some bacterial species display autofluorescence [14], potentially interfering with the detection signal. Thus, substrates that release fluorophores emitting at higher wavelengths, such as styrylcoumarin **3** [15], or those that display a significant wavelength shift before and after hydrolysis, for example 2-aminoacridone **4** [16], have potential in bacterial detection media (Scheme 1).

We based this work on the previously described chromogenic 3-hydroxyflavone (3HF) derivative **5**, which was successfully employed for the detection of β-glucosidase activity in Gram negative bacteria. Upon incorporation into a suitable culture medium and subsequent enzymatic hydrolysis by bacteria, dark brown colonies were obtained due to the chelation of the released 3HF **6** with the metal ions (Fe^3+^) present in the growth medium as a supplement (Figure 1a) [17]. Derivatives of 3HF are also known for their dual fluorescence, exhibiting emission via the excited states of both the normal **6N** (at 440 nm) and tautomer **6T** (at 524 nm) forms due to excited state proton transfer (ESIPT) (Figure 1b) [18]. The wavelength and the quantum yield of the ESIPT-derived emission peak is defined by (i) the ring substituents, and (ii) the molecular environment (e.g., solvent) of 3HF and its derivatives. Both factors contribute to the acidity of the flavone hydroxyl and/or the dipole moment of the molecule.

When the possibility for excited state intramolecular proton transfer (ESIPT) is perturbed, the tautomer is less likely to form, thus the longer wavelength emission from this form is supressed. The introduction of electron donating groups (EDG) at the para-position of the 2-phenyl substituent of 3HF—for example as in 4′-dimethylamino-3HF **7** (Scheme 2a)—results in an increased dipole moment, prompting intramolecular charge transfer (ICT) that can potentially lead to the red shift of the emission maximum and the increase of the quantum yield in polar solvents when compared to unsubstituted 3HF **6** [19,20]. In polar aprotic solvents, 3-hydroxyflavones substituted with EDGs, such as **7**, have been reported to lack the dual N* and T* emission profile and to exhibit a single emission that arises from the hydrated normal form (H-**7**N*) with a maximum red-shifted from that of the N* state (H-**7**N* at ~530–550 nm) (Scheme 2a) [21,22]. The use of probes with this optical property within complex samples offers the added benefit of a detection signal that is outwith the autofluorescence region of any microorganisms that may be present. For example, the non-fluorescent probe 4′-diethylamino-3-(2″,4″dinitrophenyl)oxyflavone **8** (Scheme 2b) was employed for thiolate sensing in aqueous media; upon thiolate-specific cleavage of the dinitrobenzyl moiety, the fluorophore emission at 538 nm was regenerated (λ_ex_ = 417 nm) [23].

In this work, 4′-dimethylamino-3HF **7** was conjugated to β-alanine via a self-immolative *p*-aminobenzyloxy linker to obtain BAP substrate **9** (Scheme 3). The substrate was incorporated into agar, which was then inoculated with 20 clinically relevant microorganisms, including the BAP producers *P. aeruginosa*, *B. cepacia*, and *S. marcescens*. Visual observation and image analysis of the colonies were then conducted in order to evaluate the performance of the substrate in the detection and identification of BAP producers.

## 2. Results and Discussion

### 2.1. Synthesis of Substrate ***9***

The chalcone precursor **10** (which was obtained via the previously reported aldol reaction between 2′-hydroxyacetophenone and 4-*N,N*-dimethylaminobenzaldehyde [24]) was cyclized using hydrogen peroxide in the presence of a base to form 4′-diethylamino-3HF **7** (Scheme 3) [24]. To obtain the linker–amino acid moiety **11**, β-alanine and the self-immolative linker precursor, 4-aminobenzylalcohol, were coupled as reported previously [15]. The mesylate of **11** was formed in situ and coupled with the fluorophore **7**, in the presence of potassium carbonate [15], to obtain the protected form of substrate **12**. The ether conjugation in **12**, which is acid labile due to the presence of the carbonyl group in the *ortho* position [25], necessitated a less acidic, two phase (ethyl acetate-water) removal of the Boc protecting group in order to obtain substrate **9** as a free amine.

### 2.2. In Vitro Fluorescence Study

The fluorescence excitation and emission spectra of fluorophore **7** and substrate **9** were recorded in a THF-water 1:1 mixture in order to solubilize the substrate while still resembling the polar protic environment of the bacterial growth media (Figure 2a, Table 1). Differences were observed in both the excitation and the emission spectra of the substrate **9** and the fluorophore **7**.

Fluorophore **7** displayed yellow-green emission, with a maximum at 536 nm upon excitation at 426 nm. A single fluorescence maximum—and the lack of the N* and T* dual emission—was observed, as expected, due to the perturbing of ESIPT by the H-bonded fluorophore in the protic solvent mixture [22]. This hindrance may also be the reason for a weaker excitation peak that is observed as a shoulder, with a maximum at 388 nm.

For the substrate **9**, ESIPT is inhibited (by design) via the substitution of the phenolic OH, disrupting the opportunity for the formation of the tautomer form, hence a more prominent normal excitation at 388 nm (with a secondary peak at 413 nm) and blue-shifted emission maximum at 517 nm was observed. The Stokes shifts of 129 and 110 nm for substrate **9** and fluorophore **7** (Table 1), respectively, are suitable for biosensing applications.

Despite the significantly overlapping emission peaks of **7** and **9**, the visually observed fluorescence (on examination with a 365 nm light source) of the respective solutions reveals a distinct difference in the emission colour (Figure 2b).

### 2.3. Biological Activity

The substrate **9** was dissolved in agar for the in-house formulation of growth media. Petri dishes were then poured and either spot inoculated with a selection of 20 clinically relevant pathogens (10 Gram-negative, 8 Gram-positive, 2 yeasts) (Figure 3a–d) or streaked with selected strains (*A. baumanii*, *E. cloacae* as negative controls, *P. aeruginosa* and *B. cepacia* as BAP producers) (Figure 3e–h). After 24 h of incubation, the plates were visually inspected under visible (Figure 3b,c) and UV light (365 nm) (Figure 3d–h) to determine inhibitory profiles and the fluorescence emitted by the colonies.

#### 2.3.1. Antimicrobial Effect

Flavonoids are known to express growth inhibitory bactericidal effects and have been suggested to result in membrane disruption and to interact with a broad range of enzymes [26]. In this study, the presence of substrate **9** in the medium had an inhibitory effect on the growth of 2 Gram-negative (*E. coli* and *K. pneumoniae*), most Gram-positive (fully inhibited: *S. pyogenes*, *Methicillin Resistant S. aureus* (MRSA), *S. aureus*, *S. epidermidis*, *B. subtilis*; malformed colonies: *L. monocytogenes*, *E. faecium*, *E. faecalis*), and the two tested yeast species (*C. albicans*, *C. glabrata*), while *A. baumanii* developed malformed colonies. The reasons for this inhibitory profile have not been investigated, and further investigation is not within the scope of this study.

#### 2.3.2. Fluorescence on Agar

The aqueous agar-based growth medium supplemented with the substrate **9** (100 mg/mL) showed no background fluorescence upon excitation at 365 nm when observed by the naked eye. This is an ideal scenario, producing minimal interference with the detection signal, resulting in increased sensitivity, and reduced detection time. 

Viable bacterial colonies, upon uptake and accumulation of the substrate **9**, displayed blue emission. This correlates with a previously reported observation in which related derivatives—monopeptides labelled with 4′-dimethylamino-3HF **7**—showed a significant increase in their fluorescence intensity upon their interaction with cell membrane [21].

Colony formation by all three BAP-producing bacteria, *S. marcescens*, *B. cepacia*, and *P. aeruginosa*, was uninterrupted, and localised yellow-green fluorescence was observed within the colonies, indicating the BAP-specific hydrolysis of the substrate **9** and good cellular adherence of the released fluorophore **7**. The observed red-shifted (blue to yellow/green) emission confirms the regeneration of the phenolic OH of the fluorophore and the prevalence of its hydrated (H-N*) form in the protic polar cellular environment. The fluorescence intensity differences observed between the tested BAP-producing bacteria are consistent with the previously reported differences in BAP expression, with *S. marcescens* displaying lower levels of BAP activity than *P. aeruginosa* [12]. 

The images obtained under observation at 365 nm were processed with ImageJ by analysing the Red Green Blue (RGB) histograms for each colony on the multispot inoculation plates (using the same size as the area of interest) (Appendix A and for the streaked colonies of *E. cloacae* and *P. aeruginosa*, respectively, and a mixed culture of these pathogens (measuring the whole plate area) to assess whether mixed cultures could be indicated using the substrate **9** (Appendix A). The mean overall intensity (Figure 4a), green (Appendix A), and blue (Appendix A) values of the respective spot-inoculated colonies of the 20 pathogens allowed for differentiation between viable and inhibited species, which is a powerful tool for inclusion in susceptibility evaluations. Moreover, the mean red values (Figure 4b) clearly indicated the strains expressing BAP activity (spot #5, 7, 9).

The intensity and respective R, G, B values generated on the whole-plate images of the streaked colonies (Appendix A) were conclusive when R/B and R/Int were calculated for mixed *E. cloacae* and *P. aeruginosa* culture (Appendix A); these values fell between the values for the respective pure streaked cultures (Appendix A). 

## 3. Materials and Methods

All solvents and reagents were purchased from Sigma-Aldrich, Castle Hill, Sydney, Australia and Alfa-Aesar (Thermo Fisher Scientific Australia Pty. Ltd.), Scoresby, Victoria, Australia and used without any further purification or treatment. Thin layer chromatography was performed on Merck 60F-254 silica gel plates. Flash chromatography was performed on GRACE Reveleris X2 (In Vitro Technologies Pty. Ltd., Noble Park, Victoria, Australia). ^1^H and ^13^C-NMR spectra were acquired on a Varian 400MR (Varian Australia Pty. Ltd., Mulgrave, Victoria, Australia) at 400 MHz and 100 MHz, respectively. Coupling constants (J) are in Hertz (Hz), chemical shifts (δ) are expressed in parts per million (ppm) and reported relative to tetramethylsilane (TMS) or relative to residual solvent peaks. Low resolution mass spectra were obtained on a Thermo Scientific TSQ Quantum Access Max LCMS/MS&TLX1 Turboflow Chromatography System (Thermo Fisher Scientific Australia Pty. Ltd.), Scoresby, Victoria, Australia) in positive ion mode.

### 3.1. Synthetic Procedures

#### 3.1.1. Preparation of 4-Dimethylamino-2′-hydroxychalcone **10**

To a solution of 2′-hydroxy acetophenone (2.65 mL, 22 mmol) and 4-*N*-dimethylaminobenzaldehyde (3.1 g, 21 mmol) in ethanol, an aqueous solution (20 mL) of KOH (3.7 g, 74 mmol) was added dropwise. The resulting solution was kept at 50 °C overnight. Upon completion, the reaction mixture was poured over water and concentrated HCl was added to reach pH 2. The precipitate formed was filtered and recrystallised from ethanol to give **10** as a red solid (2.83 g, 11 mmol, 50%). ^1^H-NMR (400 MHz, DMSO-*d*_6_) δ 3.03 (6H, s, 2 × CH_3_), 6.76 (2H, d, *J* = 8.0 Hz, 2 × CH_Ph_), 6.95–6.99 (2H, m, 2 × CH_Ar_), 7.53 (1H, td, *J* = 1.6 and 8.0 Hz, CH_Ar_), 7.73–7.78 (3H, m, 2 × CH_Ph_ and =CH), 7.83 (1H, d, *J* = 15.2 Hz, =CH), 8.26 (1H, dd, *J* = 1.6 and 8.0 Hz, CH_Ar_); ^13^C-NMR (100 MHz, DMSO-*d*_6_) δ 40.6 (2 × CH_3_), 112.2 (2 × CH_Ph_), 115.1 (=CH_α_), 118.2 (CH), 119.3 (CH), 120.9 (quat., C-2), 122.1 (quat., C-1′ or 6′), 130.9 (CH-3 or 6), 131.9 (2 × CH_Ph_), 136.2 (CH-4 or 5), 147.0 (=CH_β_), 152.8 (quat., C-4′), 162.8 (quat., C-1), 193.6 (quat., C=O); MS (ESI) *m*/*z* 268.05 [MH]^+^; HRMS *m*/*z* Found: 290.11565, calcd for C_17_H_17_NO_2_Na 290.11515.

#### 3.1.2. Preparation of **7**

To a solution of 4-dimethylamino-2′-hydroxychalcone **10** (2.5 g, 9.26 mmol) in an ethanol/THF mixture (1/1 *v*:*v*) (70/70 mL), an aqueous solution (12 mL) of NaOH (1.82 g, 45.5 mmol) was added. The resulting reaction mixture was cooled to 0 °C, and 35% aqueous H_2_O_2_ (9 mL) was added dropwise. The resulting solution was stirred at room temperature for 2 days and then poured over water (150 mL). Adjustment of the pH to 3–4 with 1N HCl resulted in a yellow precipitate upon standing overnight, this crude solid was filtered and recrystallised from ethanol to give **7** as an orange crystalline solid (2.13 g, 7.6 mmol, 82%). ^1^H-NMR (400 MHz, DMSO-*d*_6_) δ 3.02 (6H, s, 2 ×CH_3_), 6.84 (2H, d, *J* = 9.2 Hz, 2 × CH_Ph_), 7.44 (1H, t, *J* = 7.2 Hz, CH), 7.727–7.77 (2H, m, 2 × CH), 8.08 (1H, dd, *J* = 0.4 and 8.0 Hz, CH), 8.12 (2H, d, *J* = 9.2 Hz, 2 × CH_Ph_), 9.19 (1H, br s, OH); ^13^C-NMR (100 MHz, DMSO-*d*_6_) δ 40.6 (CH_3_, 2 × CH_3_), 111.8 (CH-3′ and 5′), 118.4 (CH), 118.6 (quat.), 121.9 (quat.), 124.7 (CH-6 or 7), 125.05 (CH-5 or 8), 129.4 (CH-2′ and 6′), 133.5 (CH), 137.7 (quat.), 147.25 (quat., C-1′), 151.5 (quat., C-4′), 154.7 (quat., C-4a or 8a), 172.4 (quat., C-4); MS (ESI) *m*/*z* 282.03 [MH]^+^; HRMS *m*/*z* found 304.09485 calcd for C_17_H_15_NO_3_Na 304.09441.

#### 3.1.3. Preparation of 2-(*p*-Dimethylaminophenyl)-3-[4-{3′-(*tert*-butoxycarbonylamino) Propanamido Benzyloxy]-flavone **12**

*N-*(*N*′-(*tert*-Butoxycarbonyl)-β-alanyl)-4-aminobenzyl alcohol **11** (0.58 g, 1.98 mmol) was dissolved in anhydrous DCM (20 mL) under inert atmosphere and kept in an acetone/ice bath. Then, *N,N*-diisopropylethylamine (DIPEA) (0.31 g, 2.37 mmol) was added followed by the dropwise addition of methanesulfonyl chloride (0.27 g, 2.37 mmol). The reaction mixture was stirred for 2 h at 0–5 °C. The reaction mixture was poured over an ice/conc. HCl mixture (35/15 mL), then the separated DCM layer was used without further purification in the next step assuming 100% yield of the mesylated product (based on TLC). In a separate flask, 2-4′-(dimethylamino)phenyl-3-hydroxyflavone **7** (0.4 g, 1.42 mmol) was dissolved in DCM (50 mL), and potassium carbonate (1.17 g, 7.1 mmol) was added and stirred at room temperature for 1 h. The solution of the mesylate formed previously was added dropwise, and the mixture was left to stir overnight. The organic residue was then extracted with water (100 mL), 1 M HCl (50 mL), followed by saturated brine (50 mL), then dried over Na_2_SO_4_ and concentrated under vacuum. The resulting yellow solid was recrystallised from an ethyl acetate-hexane mixture to give the crude product. Column chromatography using a gradient of 15% to 60% petroleum ether in ethyl acetate was carried out to obtain **12** as a yellow solid (0.475 g, 0.82 mmol, 58%); ^1^H-NMR (400 MHz, DMSO-*d*_6_) δ 1.37 (9H, s, C(CH_3_)_3_), 2.46 (2H, t, *J* = 7.2 Hz, CH_2β_), 3.03 (6H, s, 2 × CH_3_), 3.21 (2H, q, *J* = 6.8 Hz, CH_2α_), 5.01 (2H, s, CH_2_O), 6.81 (2H, d, *J* = 9.2 Hz, 2 × CH), 6.85 (1H, m, NH_carbamate_), 7.32 (2H, d, *J* = 8.4 Hz, 2 × CH), 7.47 (1H, t, *J* = 7.47 Hz, CH-6 or 7), 7.54 (2H, d, *J* = 8.4 Hz, 2 × CH), 7.71 (1H, d, *J*= 8.4 Hz, CH-8 or 5), 7.78 (1H, t, *J* = 7.72 Hz, CH-6 or 7), 8.02 (2H, d, *J* = 9.2 Hz, 2 × CH), 8.09 (1H, dd, *J* = 4.0 Hz and 8.0 Hz, CH-8 or 5), 9.89 (1H, s, NH_amide_); ^13^C-NMR (100 MHz, DMSO-*d*_6_) δ 28.2 (3 × CH_3_), 36.5 (CH_2β_), 36.75 (CH_2α_), 39.6 (2 × CH_3_), 72.40 (OCH_2_), 77.6 (quat., C(CH_3_)_3_), 111.2 (CH-3′ and 5′), 116.70 (quat., C-1′), 118.14 (CH-8), 118.70 (CH-3″ and 5″), 123.5 (quat., C-4a), 124.75 (CH-5 or 6), 124.81 (CH-5 or 6), 129.0 (CH-2″ and 6″), 129.7 (CH-2′ and 6′), 131.4 (quat., C-1″), 133.53 (CH-7), 137.7 (quat., C-3), 139.0 (quat., C-4″), 151.7 (quat., C-4′), 154.4 (quat., C-8a), 155.5 (quat., C=O_carbamate_), 156.2 (quat., C-2), 169.4 (quat., C=O_amide_), 175.3 (quat., C=O, C-4); MS (ESI) *m*/*z* 558.16 [MH]^+^; HRMS Found 580.24213 calcd for C_32_H_35_N_3_O_6_Na 580.24181.

#### 3.1.4. Preparation of 2-(*p*-Dimethylaminophenyl)-3-[4-propanamido benzyloxy]-flavone **9**

2-(*p*-dimethylaminophenyl)-3-[4-{3′-(*tert*-butoxycarbonylamino)propanamidobenzyloxy]-flavone **12** (250 mg, 0.45 mmol) was dissolved in an ethyl acetate (10 mL) and DMF (2 mL) mixture, then 3 M aqueous HCl (10 mL) was added. The resulting reaction mixture was stirred at room temperature for 3 h. To the separated aqueous layer, ethyl acetate was added, and the pH of this stirred mixture was adjusted to 7 using 5 M aqueous NaOH. The precipitate formed was filtered and recrystallized from ethyl acetate to give **9** as a yellow solid (108 mg, 0.24 mmol, 53%). ^1^H-NMR (400 MHz, DMSO-*d*_6_) δ 2.42 (2H, t, *J* = 6.4 Hz, CH_2α_), 2.86 (2H, m, CH_2α_), 3.03 (6H, s, 2 × CH_3_), 5.01 (2H, s, CH_2_O), 6.82 (2H, d, *J* = 9.3 Hz, 2 × CH), 7.33 (2H, d, *J* = 8.4 Hz, 2 × CH), 7.47 (1H, t, *J* = 7.5 Hz, CH-6 or 7), 7.55 (2H, d, *J* = 8.4 Hz, 2 × CH), 7.72 (1H, d, *J* = 7.8 Hz, CH-8 or 5), 7.79 (1H, t, *J* = 7.8 Hz, CH-6 or 7), 8.03 (2H, d, *J* = 9.2 Hz, 2 × CH), 8.09 (1H, dd, *J* = 4.0 and 8.0 Hz, CH-8 or 5), 10.11 (1H, s, NH_amide_); ^13^C-NMR (100 MHz, DMSO-*d*_6_) δ 35.1 (CH_2β_), 35.8 (CH_2α_), 39.6 (2 x CH_3_), 72.40 (OCH_2_), 111.2 (CH-3′ and 5′), 116.7 (quat., C-1′), 118.1 (CH-8), 118.8 (CH-3″ and 5″), 123.5 (quat., C-4a), 124.7 (CH-5 or 6), 124.8 (CH-5 or 6), 129.0 (CH-2″ and 6″), 129.7 (CH-2′ and 6′), 131.65 (quat., C-1″), 133.5 (CH-7), 137.7 (quat., C-3), 138.7 (quat., C-4″), 151.7 (quat., C-4′), 154.4 (quat., C-8a), 156.2 (quat., C-2), 168.5 (quat., NHC=O), 173.3 (quat., C=O, C-4); MS (ESI) *m*/*z* 458.05 [MH]^+^; HRMS found 480.18987 calcd for C_27_H_27_N_3_O_4_Na 480.18938.

### 3.2. In Vitro Fluorescence Studies

The fluorescence of 0.01 mg/mL aqueous THF (1:1 *v*:*v*) solutions of the substrate and the fluorophore was evaluated with 5 nm slit width at low sensitivity, with 1 nm steps for emission and excitation, respectively.

### 3.3. Biological Evaluation

#### 3.3.1. Preparation of Culture Media Containing Substrate **9**

Columbia agar was prepared as follows: 41 g of Columbia agar (Oxoid Basingstoke, UK) was added to deionized water, and the volume was made up to 1 L. The medium was sterilized by autoclaving at 116 °C for 20 min and left to cool at 50 °C. 2 mg of the substrate to be tested was initially dissolved in 100 μL of *N*-methylpyrrolidone, and this was added to Columbia agar (made up to 20 mL) then poured into sterile Petri dishes to give a final concentration of 100 mg/L for the substrate **9**. Columbia agar incorporating an equivalent concentration of *N*-methylpyrrolidone was used as a growth control.

#### 3.3.2. Microbial Suspension Preparation

Microbial reference strains were obtained from either the National Collection of Type Cultures (NCTC) or the National Collection of Pathogenic Fungi (NCPF), which are both located at the Central Public Health England Laboratory, Colindale, UK or the American Type Culture Collection (ATCC), Manassas, USA. The 20 test microorganisms were maintained on Columbia agar.

#### 3.3.3. Multipoint Inoculation

Colonies of each microbial strain were harvested using a loop from overnight cultures on Columbia agar. These were suspended in sterile deionized water to a suspension equivalent to 0.5 McFarland units using a densitometer. 100 μL of this suspension was pipetted into the corresponding wells of a multipoint inoculation device. Each set of plates received 1 μL of bacterial suspension, giving 1.5 × 10^5^ organisms per spot on each inoculation. Twenty strains were inoculated per plate, and the plates were incubated for 18 h in air at 37 °C.

#### 3.3.4. Activity Determination

After incubation, the activity of the microorganisms with the test substrates was determined by observing the plates under UV irradiation at 365 nm and comparing with the substrate-free control.

## 4. Conclusions

In this work, a novel 3-hydroxyflavone-based β-alanine aminopeptidase substrate was designed, synthesised, characterised for its chemical and optical properties, and evaluated on an agar growth medium for its potential use as a selective fluorescent probe for the detection of BAP-producing bacteria. The fluorescence characteristics in aqueous solution were as expected. Conversion of the fluorophore **7** into substrate **9** resulted in the hindrance of ESIPT and a change in the emission wavelength. The agar-based growth medium supplemented with the substrate showed no background fluorescence, which is desirable for this application. Upon inoculation with the selected pathogens, growth inhibition was observed for some of the species; however, the microorganisms of interest formed viable, fully grown colonies. Viable colonies displayed blue fluorescence upon the accumulation of the substrate, confirming the cell-wall permeability of the substrate and this, as an added benefit, generally indicating viable bacteria. More significantly, BAP-producing bacteria displayed green fluorescence upon selective hydrolysis of the substrate. This fluorescence was localised within the colonies, which is beneficial for their identification. The change in fluorescence emission upon enzymatic cleavage was readily observed under a UV lamp, while a simple-to-conduct image analysis allowed the differentiation of BAP producers. The simultaneous detection and differentiation of viable colonies and BAP producers (while displaying no background fluorescence) achieved using substrate **9** is a desirable advancement to the previously reported fluorogenic BAP substrates.

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
