# Peer review of "A Selective, Dual Emission β-Alanine Aminopeptidase Activated Fluorescent Probe for the Detection of Pseudomonas aeruginosa, Burkholderia cepacia, and Serratia marcescens"

_molecules, 2019, doi:10.3390/molecules24193550_

Round 1

Reviewer 1 Report

In this manuscript, the authors reported the design, synthesis and characterization of a novel 3-hydroxyflavone-based β-alanine aminopeptidase. They have shown the fluorescent probe that was applicable for the detection of BAP producing bacteria. The concept of this work is great and through the improvement of the following minor points, can be published.

Minor Comments:

First page line 93, “….For example, use of colourimetric BAP substrates, e.g. 7-amino-1-36 pentyl-3H-phenoxazin-3-one 1 (1-pentylresorufamine [PRF]) (Scheme 1) [9]…

Clarify the “REF”

Scheme 3, the compound 11 that undergoes the methanesulfonyl chloride reaction; yield should be added. Figure 2, it is recommended that the chemical structure of compounds 9 and 7 to  be added to fluorescence spectra / solutions with correlation with the blue and green colour.   Figure 3, (d) under UV light; add the range and lambda max of UV excitation. Is it 365 nm?

Author Response

Dear Mr Senk,

Manuscript ID: molecules-602588

A selective, dual emission β-alanine aminopeptidase activated fluorescent probe for the detection of Pseudomonas aeruginosa, Burkholderia cepacia, and Serratia marcescens

Thank you for your consideration and the reviewers’ comments. Here, we would like to address the comments, and attach our revised manuscript.

Reviewer #1

First page line 93, “….For example, use of colourimetric BAP substrates, e.g. 7-amino-1-36 pentyl-3H-phenoxazin-3-one 1 (1-pentylresorufamine [PRF]) (Scheme 1) [9]… Clarify the “REF”

We have amended the sentence for better clarity. Please see revised manuscript.

Scheme 3, the compound 11 that undergoes the methanesulfonyl chloride reaction; yield should be added.

Thank you, the change has been made. Please see revised manuscript.

Figure 2, it is recommended that the chemical structure of compounds 9 and 7 to be added to fluorescence spectra / solutions with correlation with the blue and green colour.

Thank you, we modified Figure 2 according to your suggestion. Please see revised manuscript.

Figure 3, (d) under UV light; add the range and lambda max of UV excitation. Is it 365 nm?

Yes, thank you. We have added the wavelength to the legend. Please see revised manuscript.

Reviewer #2

The novelty of this work needs to be clarified. The authors did not describe the advances between their probe and those reported in the literature in the Introduction or elsewhere in the main text.

Thank you for pointing this out. We have added a sentence to the conclusion to summarise the novelty of our method. Please see revised manuscript.

The author claimed that "use of colourimetric BAP substrates...can result in up to 99.9% ... however, the time taken to obtain reliable results can be 18-72 hours". The reviewer is curious about the selectivity and time spend to obtain reliable results.

The colourimetric substrate was extensively tested against different strains of bacteria in the reference cited in order to determine the sensitivity and selectivity of the method. The results in the cited reference show that detection of P. aeruginosa on the basis of BAP activity is both sensitive (99% P. aeruginosa strains hydrolyse the substrate) and selective (substrate only hydrolysed by BAP producers); we chose not to include a more detailed explanation in the current manuscript.

The author should supply data (NMR) to support the mechanism that 9 can change to 7 after BAP activity.

The formation of the fluorophore 7 was confirmed by analogy to all published work and the generation of the expected fluorescence correlating with the in vitro fluorescence produced by both isolated fluorophore 7 and substrate 9. In addition, it would not really be feasible to isolate sufficient quantities of the hydrolysis product, the fluorophore 7, from the bacterial growth medium in order to obtain an NMR spectrum.

We hope these responses are satisfactory.

With best wishes.

Yours sincerely

Linda Váradi MSc, PhD, MRSC

Vice Chancellor’s Postdoc Fellow

School of Engineering

RMIT University

E [email protected]

Reviewer 2 Report

In the manuscript, the author designed a 3-hydroxyflavone-based β-alanine aminopeptidase substrate for the detection of BAP producing bacteria. The manuscript was concisely written and the conclusion was supported with experimental data. Therefore, the reviewer recommends this manuscript for the publication after the revision of a following point.

The novelty of this work needs to be clarified. The authors did not describe the advances between their probe and those reported in the literature in the Introduction or elsewhere in the main text. The author claimed that "use of colourimetric BAP substrates...can result in up to 99.9% ... however, the time taken to obtain reliable results can be 18-72 hours". The reviewer is curious about the selectivity and time spend to obtain reliable results. The author should supply data (NMR) to support the mechanism that 9 can change to 7 after BAP activity.

Author Response

(The authors gave the same response as above.)
